# Usefulness of Adding Maspin Staining to p53 Staining for EUS-FNA Specimens of Pancreatic Ductal Adenocarcinoma

**DOI:** 10.3390/jcm11206097

**Published:** 2022-10-16

**Authors:** Koh Fukushi, Akira Yamamiya, Keiichi Tominaga, Yoko Abe, Koki Hoshi, Kazunori Nagashima, Ken Kashima, Yasuhito Kunogi, Fumi Sakuma, Hidetsugu Yamagishi, Kazuyuki Ishida, Yasuo Haruyama, Atsushi Irisawa

**Affiliations:** 1Department of Gastroenterology, Dokkyo Medical University, Mibu 321-0293, Japan; 2Academic Institutional Research Center, Dokkyo Medical University, Mibu 321-0293, Japan; 3Department of Diagnostic Pathology, Dokkyo Medical University, Mibu 321-0293, Japan; 4Integrated Research Faculty for Advanced Medical Science, Dokkyo Medical University, Mibu 321-0293, Japan

**Keywords:** maspin, pancreatic ductal carcinoma, p53, EUS-FNA, immunohistochemical

## Abstract

Background: Endoscopic ultrasound-guided puncture aspiration biopsy (EUS-FNA) of pancreatic ductal adenocarcinoma (PDAC) is highly diagnostic, but it is difficult to distinguish from benign disease. Our objective was to determine the usefulness of maspin staining, in addition to conventional p53 staining, in the diagnosis of PDAC by EUS-FNA. Methods: Of the patients who underwent EUS-FNA and were diagnosed with PDAC, we retrospectively identified 90 cases in which both maspin and p53 staining were performed. In addition, we identified 28 cases of benign pancreatic disease diagnosed using EUS-FNA and these were selected as a control group. For analysis of EUS-FNA specimens, Cohen’s Kappa (κ) coefficient and the prevalence and bias adjusted Kappa statistic (PABAK) were applied to assess the significance of sensitivity and specificity, comparing p53, maspin, p53+maspin. Results: The sensitivity and specificity of p53 staining were 48.9% and 100%. The κ coefficient was 0.31 (95%CI 0.18–0.44) (*p* < 0.01) and the PABAK coefficient was 0.22 (95%CI 0.03–0.40). The results for maspin staining were 88.9% and 92.9%. The κ coefficient was 0.72 (95%CI 0.54–0.90) (*p* < 0.01) and the PABAK coefficient was 0.78 (95%CI 0.64–0.88). The results for the combination of maspin and p53 staining were 94.4% and 92.2%. The κ coefficient was 0.82 (95%CI 0.64–1.00) (*p* < 0.01) and the PABAK coefficient was 0.86 (95%CI 0.74–0.94). Conclusion: Adding maspin staining to p53 staining showed high sensitivity and specificity. Our results demonstrated the usefulness of their combined use that might contribute to the improvement of tissue diagnostic performance of PDAC by EUS-FNA.

## 1. Introduction

In 1992, endoscopic ultrasound-guided fine-needle aspiration biopsy (EUS-FNA) was clinically applied for the first time [1]. Since then, it has become extremely popular because of its high diagnostic ability and safety. The diagnostic ability of EUS-FNA for pancreatic solid masses has been shown to be high, with a sensitivity of 86.8% and a specificity of 95.8%, in a recent systematic review [2]. Moreover, EUS-FNA for pancreatic ductal adenocarcinoma (PDAC) shows high specificity. Although it also has high sensitivity and is superior to pancreatic juice and brushing cytology [3,4], it varies from 73.9% to 96.6% [5,6,7,8,9,10,11]. Pathological diagnosis using immunohistochemical staining is useful as a method for improving the accuracy of discrimination between benign and malignant pancreatic diseases and for improving the diagnosis rate [12]. As for the use of immunohistochemical staining in the diagnosis of PDAC, p53 staining has good cancer specificity and is observed frequently [13,14]. In addition, the staining rate is low for benign diseases, such as chronic pancreatitis. Therefore, it is used routinely and traditionally. However, p53 staining presents the difficulty that it is often negative for poorly differentiated tumors and small sample volumes. The positive rate of PDAC in EUS-FNA specimens is reported as 50–70% [15,16,17]. Therefore, immunohistochemical staining with p53 alone might not be sufficient as an auxiliary diagnosis.

Recently, maspin, which is a tumor suppressor gene identified in the normal mammary gland and a member of the serpin protein family [18], was shown to be frequently expressed in cases of pancreatic cancer [19]. Maspin is not expressed in a normal pancreas [19,20], but is expressed with canceration [19]. Because the rate of diagnosis of EUS-FNA in PDAC must be improved and differentiated from those of benign diseases, adding maspin staining to the p53 staining that is performed on a daily basis might engender improvement in the rate of PDAC diagnosis. This study was conducted to clarify the additional effects of maspin staining in addition to conventional p53 staining for the diagnosis of PDAC by EUS-FNA.

## 2. Materials and Methods

### 2.1. Study Design

This single-center retrospective study investigated the usefulness of immunohistochemical staining of maspin and p53 expression in specimens of PDAC obtained by EUS-FNA. The primary endpoint was the evaluation of the improvements in sensitivity and specificity by adding maspin immunostaining to p53 for the diagnosis of PDAC. The secondary endpoint was the evaluation of the relation between maspin expression and the clinical factors, and whether maspin can be regarded as a predictive indicator for prognosis.

This study was reviewed and approved by the Institutional Review Board of Dokkyo Medical University (approval no. R-24-6J), conducted in accordance with the human and ethical principles of research set forth in the Declaration of Helsinki. This study was registered with the University Hospital Medical Information Network registration number UMIN000048015. A means to opt out was provided to participants instead of informed consent, which is one means of guaranteeing opportunities for research subjects to be notified and to support the publishing of research information on our website.

### 2.2. Patients/Study Population

A total of 191 consecutive cases were selected. For all the cases, EUS-FNA had been performed at Dokkyo Medical University Hospital between April 2012 and December 2017 and a definitive diagnosis of PDAC had been obtained, with sufficient specimens for histological evaluation. The exclusion criteria were the following: (a) cases in which either or both maspin staining and p53 staining had not been performed, (b) patients with no prognostic follow-up as of 31 March 2018, (c) patients with PDAC in the remnant pancreas, (d) patients with radiotherapy or chemotherapy before EUS-FNA, (e) and patients with missing data. The remaining 90 patients were evaluated (Figure 1). The definitive diagnosis of PDAC was based on pathological diagnosis of surgical specimens or, in advanced cases, histological diagnosis with EUS-FNA specimens, imaging, blood and biochemical tests, and the clinical course.

In addition, 44 consecutive cases of benign pancreatic disease, diagnosed using EUS-FNA, of pancreatic masses during the same period were selected. Among these were 28 cases with good specimens that were suitable for immunohistochemical staining. These cases were selected as the control group; p53 and maspin staining were applied to them. Benign disease was evaluated by regular outpatient follow-up for at least 12 months after EUS-FNA to confirm through imaging studies that the disease was not progressive.

### 2.3. Procedure and Intervention

EUS-FNA was performed using a curved linear array echoendoscope (GF-UCT260; Olympus Corp., Tokyo, Japan) and needles (Acquire; Boston Scientific Japan, Tokyo, Japan or EchoTipUltra; Cook Scientific Japan, Tokyo, Japan, or EchoTip ProCore; Cook Scientific Japan, Tokyo, Japan). All procedures were performed at the participating facility by an experienced endosonographer (>100 EUS-FNA procedures) or trainee under the supervision of experts. After intravenous injection of midazolam and pentazocine, EUS-FNA was performed in the left lateral decubitus position under moderate conscious sedation.

Once the needle had been placed clearly in the target lesion, the stylet was withdrawn completely. Then, the needle was moved back and forth approximately 10–20 times within the lesion under EUS guidance, while suction was applied using a 10 or 20 mL syringe with a lock device. If blood contamination was obvious, then the slow pull technique was used. The material obtained when using EUS-FNA was placed on glass slides or placed in a container by pushing air from a syringe and/or pushing the stylet, and the materials were processed for cytological and histological examination. Most of the tissue specimens were fixed immediately in 10% neutral-buffered formalin solution for histological examination. The other specimens were used for rapid onsite evaluation. When the cytopathologist confirmed that sufficient material had been obtained, EUS-FNA was finished.

### 2.4. Histological Evaluation

Specimens for histology were stained with hematoxylin and eosin (H&E), and additional immunohistochemistry using maspin and p53 staining. PDAC and benign pancreatic disease were diagnosed based on H&E staining. For this study, pathological diagnosis using H&E staining was performed in real time by several pathologists. The immunohistochemical staining was performed in real time, and the evaluation of maspin/p53 staining was performed at a later date by one pathologist (H.Y.) who was blinded to the final diagnosis. 

The degree of pathological differentiation was evaluated according to the World Organization of Health classification.

### 2.5. Evaluation of Immunohistochemical Staining

In maspin staining, an appropriate cut off value (%) was set using the ROC curve. Positive classification was inferred when the nucleus or cytoplasm of tumor cells above the cut off value (%) was stained (Figure 2). With reference to past literature [21,22,23], p53 staining was inferred as positive when 60% or more of the tumor cell nuclei was stained. Sensitivity and specificity were analyzed for (1) the samples with only p53 staining, (2) only maspin staining, and (3) p53 staining + maspin staining (the test was considered positive if any marker was positive).

### 2.6. Investigation of the Patient Prognosis

The patients were managed using surgery, chemotherapy, and follow-up without treatment. The prognosis for each patient was analyzed based on medical records, and was evaluated by the kind of management after diagnosis. All patients had follow-ups until their death or until their most recent contact or visit (through 31 March 2018).

### 2.7. Statistical Analysis

Statistical analyses were conducted using various softwares (SPSS ver. 21; IBM-SPSS, Inc. Chicago, IL, USA and ver. 4.2.0, R software). Quantitative data are presented as the median (range). For specimens obtained using EUS-FNA, the p53 and maspin positive rates were summarized as numbers and percentages. We used the receiver operating characteristics (ROC) curve to determine an appropriate cut-off line of maspin staining to discriminate between PDAC and benign. For analysis of EUS-FNA specimens, Cohen’s Kappa (κ) coefficient was applied to assess the significance of sensitivity, and specificity, comparing p53, maspin, p53+maspin. Regarding agreement, κ values of <0 were considered to have poor agreement, with 0.01–0.20 signifying slight agreement, 0.21–0.40 fair agreement, 0.41–0.60 moderate agreement, 0.61–0.80 substantial agreement, and 0.81–1.0 almost perfect agreement. Because of the presence of the bias, the prevalence and bias adjusted Kappa statistic (PABAK) was also calculated. Survival analysis was performed based on maspin expression. Clinicopathological factors (including age, sex, tumor location, degree of pathological differentiation, biomarker, clinical stage and treatment) were compared using the chi-square test, Fisher’s tests, and *t* test. The duration of overall survival (OS) was defined as the time from the initial day of obtaining a definitive diagnosis to the date of death or their most recent contact or visit (through 31 March 2018). Survival distributions were estimated using the Kaplan–Meier method. The significance of differences between survival rates was assessed using the log-rank test. For the maspin-positive and maspin-negative groups, OS was compared using Cox proportional hazards regression, from which hazard ratios (HRs) were estimated. Probability (*p*) values of < 0.05 were inferred as significant.

## 3. Results

### 3.1. Background of the Patients

Table 1 presents the patients’ background information. There were 90 PDAC cases (male:female ratio of 1.9:1) with a median age of 71 years (range, 42–90). In addition, there were 28 cases of benign disease (male:female ratio of 1.8:1) with a median age of 69 years (range, 37–85). The breakdown indicated that 18 patients had autoimmune pancreatitis, 7 patients had inflammatory pancreatic masses, and 3 patients had chronic pancreatitis.

### 3.2. Setting the Cut-Off Value of Maspin

The cut-off value of the maspin staining rate was calculated using the ROC curve (Figure 3). The best cut-off value was 11.685%. Using this cut-off value, the positive rates for PDAC were as follows: p53: 48.9%, maspin: 88.9%l maspin + p53: 94.4%. The positive rates for benign pancreatic disease were as follows: p53: 0%, maspin: 10.7%; maspin + p53: 10.7%. Table 2 presents the results.

### 3.3. Examination of the Maspin Additive Effect

The sensitivity and specificity of p53 staining were 48.9% and 100%, respectively. Regarding agreement, the κ coefficient was 0.31 (95%CI 0.18–0.44) (*p* < 0.01); the PABAK coefficient was 0.22 (95%CI 0.03–0.40). The sensitivity and specificity of maspin staining were 88.9% and 92.9%, respectively. Regarding agreement, the κ coefficient was 0.72 (95%CI 0.54–0.90) (*p* < 0.01); the PABAK coefficient was 0.78 (95%CI 0.64–0.88). The sensitivity and specificity of the combination of maspin and p53 staining were 94.4% and 92.2%, respectively. Regarding agreement, the κ coefficient was 0.82 (95%CI 0.64–1.00) (*p* < 0.01); the PABAK coefficient was 0.86 (95%CI 0.74–0.94). The results are presented in Table 3. In addition, the sensitivity and specificity of H&E were 97.8% and 100%, 100% and 100% for H&E adding maspin staining and 97.8% and 100% for H&E plus p53 staining, respectively.

### 3.4. Clinicopathologic Study of Maspin-positive and Maspin-Negative Patients

Compared to the maspin-positive group, the maspin-negative group had a larger tumor size (*p* = 0.008), more poorly differentiated tumors (*p* = 0.033), and significantly more distant metastases (*p* = 0.047) (Table 4). Cumulative survival was significantly shorter in the negative group (*p* = 0.006) (Figure 4).

## 4. Discussion

The strength of this study is that it is the first reported study that demonstrates that performing both p53 staining and maspin staining for PDAC is useful for diagnosis. The addition of maspin staining to p53 staining provided excellent results, with sensitivity and specificity values of 94.4% and 92.2%. Many additional effects of immunohistochemical staining in PDAC have been reported. Of these, p53 is found in about 50–70% of PDAC [15,16,17], but it is insufficient as an auxiliary diagnosis in EUS-FNA specimens. In addition, the respective utilities of pVHL [24], cytokeratin 17 [25], S100P [26], IMP3 [27], and mesothelin [28] have been reported. Nash et al. reported the maspin positivity rate for EUS-FNA specimens of PDAC as 91% [29]. Other reports have described that maspin staining is useful for differentiating PDAC from benign disease [30,31,32]. In this study, p53 staining alone was found to have a sensitivity of 48.9% and specificity of 100%. In addition, maspin staining alone was found to have a sensitivity of 88.9% and specificity of 92.9%. Consistent with known reports, these results are favorable even when used alone. Furthermore, as a result of the analysis of the additional effect of maspin staining on p53 staining, better agreement (substantial agreement) was shown for maspin staining compared to p53. A higher degree of agreement (almost complete agreement) was observed by adding maspin to p53. The combination of these two stains might contribute to improving the diagnosis rate of PDAC for EUS-FNA specimens. Liu et al. [33] reported some immunostaining combinations used for the diagnosis of PDAC. They found that the combination of pVHL, Maspin, S100P, and IMP-3 was the best among many combinations. The use of various modes of immunohistochemical staining is beneficial for improving the diagnostic yield. Nevertheless, using many immunohistochemical staining modes routinely in daily practice is difficult. In daily practice, p53 is used frequently, particularly for various cancers. Adding only maspin staining to p53 staining has proven to be useful in improving the diagnosis rate of PDAC.

Reportedly, maspin is not expressed in a normal pancreas; maspin is expressed with canceration [19]. Although maspin is expressed in a normal mammary gland and prostate gland, its expression is suppressed with the progression and malignancy of cancer [34,35]. The expression pattern might differ depending on the tumor type. Fitzgerald et al. [20] concluded that the expression of maspin in PDAC is mediated by demethylation of the maspin promoter and hyperacetylation of related histones, which leads to maspin expression via epigenetic depression of the maspin locus. It is responsible for tumor growth. The results of this study indicate that the positive rate of maspin staining in benign diseases was 10.7% and 88.9% in PDAC, showing that maspin is highly expressed in PDAC, which is similar to earlier reports. Maspin is also expressed in PanIN-1 [36], suggesting that hypomethylation of the maspin promoter might occur in the early stage of carcinogenesis [20,37]. If maspin is expressed in the early stage of PDAC, then maspin staining might be the key to early detection of PDAC. Guerra et al. [38] reported that an active mutation of the K-ras gene and subsequently acquired expression of a loss-of-function mutant p53 because of inflammation are necessary for pancreatic carcinogenesis. Furthermore, it is possible that p53 is not expressed in the early stage [39,40,41]. Therefore, the combination of maspin and p53 might contribute to diagnosis of various stages of carcinogenesis.

This study found that the maspin negative group had a significantly higher proportion of poorly differentiated, distant metastases, in addition to poorer prognosis. Several reports have been published regarding prognosis. Cao et al. [36] and Uchinaka et al. [42] reported that the prognosis was poor in maspin-positive cases. However, in this study, the maspin-positive cases had a relatively good prognosis. These reports [36,42] only examined patients who were eligible for surgery, whereas this study included unresectable cases, and most of the maspin-negative cases were Stage IV. This difference in patient background may have led to the discrepancy in the results. On the other hand, some reports [18,35] have suggested that the function of maspin in vitro is to inhibit migration and proliferation of tumor cells during tumor invasion and metastasis, and that maspin expression in PDAC might work in a tumor-suppressive manner. Similar to this study, a report [43] that describes a study that examined the clinicopathological significance of maspin staining of PDAC reported that the high-expression group of maspin had significantly more low-grade cancers than the low-expressing group. When the expression of maspin is low, the tumor suppressive effect of maspin is weakened. As a result, these findings suggest that the cancer progresses and metastasizes more progressively. The results of this study imply that maspin also acts on PDAC in a tumor-suppressive manner. More studies are needed to investigate the relationship between the degree of cancer progression and the expression of maspin. In addition, the previous papers are retrospective studies and larger prospective studies are needed to show anything definite.

Although immunohistochemical staining markers are useful for diagnosing PDAC, no completely cancer-specific marker exists. The same is true for maspin staining. Hence, it is important to highlight the pitfalls of the respective markers. First, the criteria for positive maspin staining depend on the report. Therefore, judgments might differ depending on the pathologist. For this study, the ROC curve was used to define the cut-off value, which might have caused a more accurate diagnosis rate. Second, the sensitivity of the maspin staining was very high in this study, but even benign disease tended to be stained slightly. Similarly, Nash et al. reported that chronic pancreatitis was also stained in 29% of cases and that it was partially weakly stained [29]. In addition, when EUS-FNA specimens are contaminated with gastrointestinal mucosal epithelium, especially when regenerative changes are added to the orbital epithelium, the specimens might resemble PDAC with weak morphological atypia, leading to false positive maspin staining and decreased specificity. Needles have been improved to prevent contamination of the gastrointestinal mucosa. The use of a stylet might prevent such contamination.

This study has several limitations. First, it was conducted as a single-center retrospective study. Selection bias is undeniable. For example, in some cases, mapsin staining was not performed if there was a low probability of PDAC in H&E. Second, the EUS-FNA puncture method and needle were not standardized. Third, the gold standard for definitive diagnosis might vary between surgical specimens and EUS-FNA tissue specimens, but the clinical course is considered to support the histopathological diagnosis. Its influence is assumed to be small.

In conclusion, the combination of maspin and p53 immunohistochemical staining showed high sensitivity and specificity. Their combined use might contribute to the improvement of tissue diagnostic performance of PDAC by EUS-FNA.

## Figures and Tables

**Figure 1 jcm-11-06097-f001:**
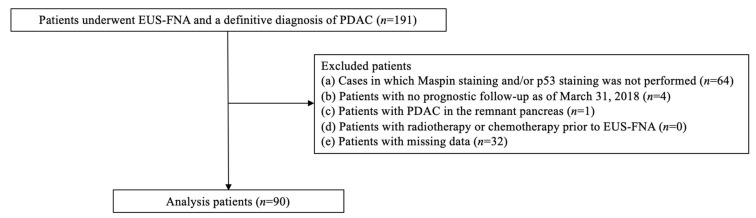
Flow chart of this study. EUS-FNA: endoscopic ultrasound-guided fine-needle aspiration biopsy; PDAC, pancreatic ductal adenocarcinoma.

**Figure 2 jcm-11-06097-f002:**
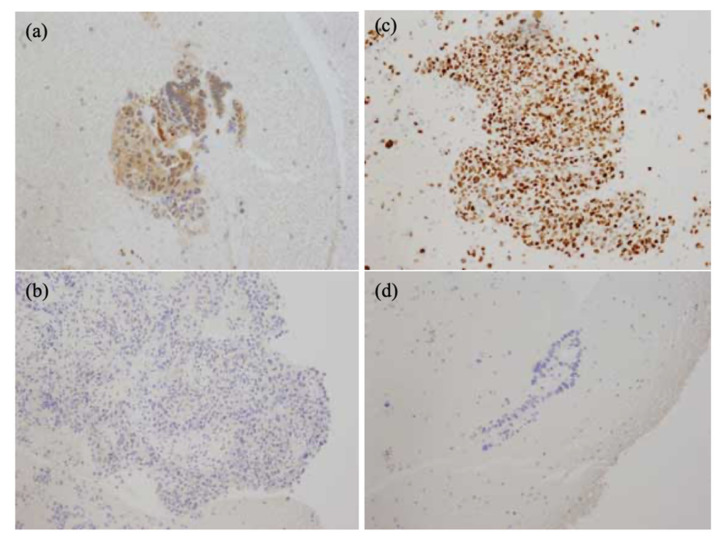
Representative positive and negative case for maspin and p53 staining. (**a**): positive for maspin staining (nucleus or cytoplasm is stained), (**b**): negative for maspin staining, (**c**): positive for p53 staining (nucleus is stained), (**d**): negative for p53 staining.

**Figure 3 jcm-11-06097-f003:**
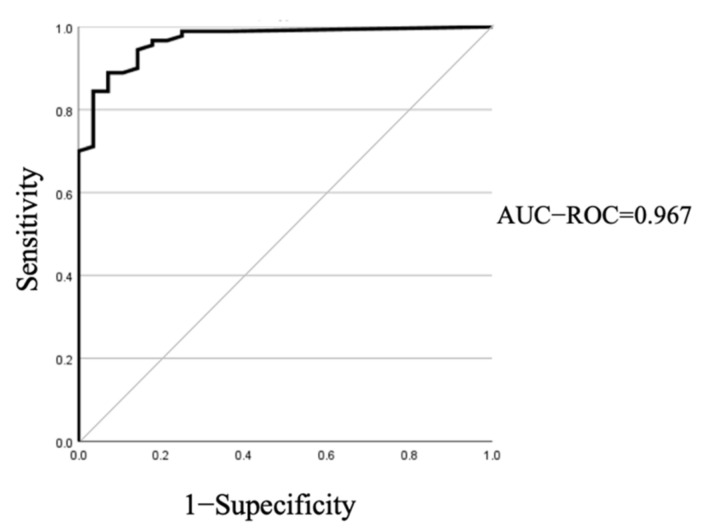
The receiver operating characteristics (ROC) curve used to determine an appropriate cut-off line of maspin staining. AUC: Area under the curve.

**Figure 4 jcm-11-06097-f004:**
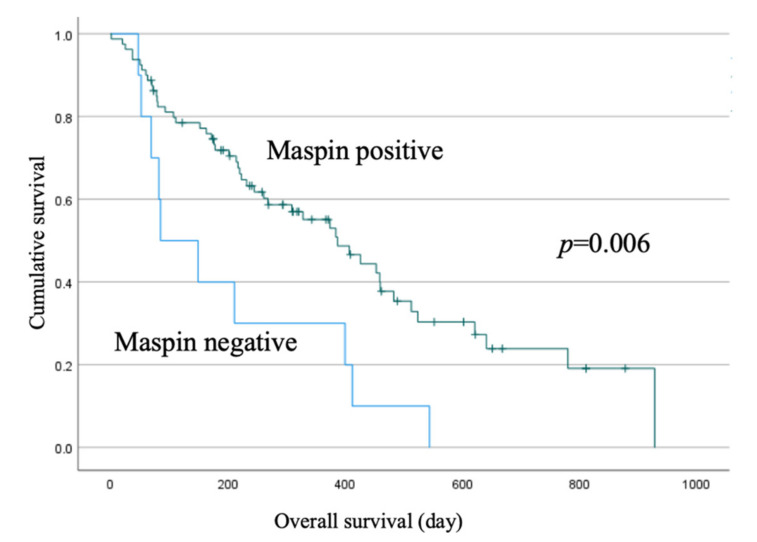
Kaplan–Meier curves of overall survival (OS) in patients with pancreatic cancer in the maspin-positive (*n* = 80) vs. maspin-negative (*n* = 10) group.

**Table 1 jcm-11-06097-t001:** Baseline characteristics of the 90 patients of PDAC.

Male/Female (*n* = 90)	66/34
Age [median (range)]	71 (42–90)
Tumor size (mm) [median (range)]	29.9 (10–64)
Location of pancreatic lesion (Uncinate process/head/body/tail)	11/42/26/11
Tumor marker [median (range)]	
CA19-9 (U/mL)	526.5 (2–12,000)
CEA (ng/mL)	4.1 (0.6–750)
DUPAN2 (U/mL)	370 (25–280,000)
SPAN1 (U/mL)	100 (5–90,000)
TNM-Staging	
T1/T2/T3/T4	11/13/42/24
N0/N1	52/38
M0/M1	46/44
Stage	
I/II/III/IV	11/21/14/44

**Table 2 jcm-11-06097-t002:** The positive rate for PDAC and benign pancreatic disease of immunohistochemical staining.

	PDAC (*n* = 90)	Benign (*n* = 28)
p53	44 (48.9%)	0 (0%)
Maspin	80 (88.9%)	3 (10.7%)
p53+maspin	85 (94.4%)	3 (10.7%)

PDAC, pancreatic ductal adenocarcinoma.

**Table 3 jcm-11-06097-t003:** The sensitivity, specificity, PPV, NPV, Youden index, κ, PABAK of H&E, p53 and maspin, either alone or in combination, for the diagnosis of PDAC.

Marker	Sensitivity (%)	Specificity (%)	PPV (%)	NPV (%)	Youden	Cohen’s Kappa (κ) Coefficient (95%CI)	*p*-Value ^†^	PABAK (95%CI)
H&E	97.8	100	100	93.3	0.978	0.954 (0.774–1.135)	<0.001	0.966 (0.880–0.996)
p53	48.9	100	100	62.1	0.489	0.312 (0.181–0.443)	<0.001	0.220 (0.032–0.397)
Maspin	88.9	92.9	97.5	72.2	0.818	0.720 (0.541–0.898)	<0.001	0.780 (0.638–0.880)
Maspin + p53	94.4	92.9	97.7	83.9	0.873	0.817 (0.637–0.997)	<0.001	0.864 (0.742–0.941)
H&E +p53	97.8	100	100	93.3	0.978	0.954 (0.774–1.135)	<0.001	0.966 (0.880–0.996)
H&E + Maspin	100	100	100	100	1.000	1.000 (0.820–1.180)	<0.001	1.000 (0.938–1.00)
H&E + Maspin +p53	100	100	100	100	1.000	1.000 (0.820–1.180)	<0.001	1.000 (0.938–1.00)

^†^ Using the κ coefficient. PPV, positive predictive value; NPV, negative predictive value; PABAK, the prevalence and bias adjusted Kappa statistic.

**Table 4 jcm-11-06097-t004:** Comparison of clinicopathologic factors between patients whose tumors showed expression of maspin and patients whose tumors showed no expression of maspin.

Clinicopathological Factors	Maspin Status		*p*-Value ^†^
	Maspin Positive (*n* = 80)	Maspin Negative (*n* = 10)	
Sex (M:F)	50	6	1.00
Age [median (range)]	70	69.6	0.912
Tumor size (mm) [median (range)]	28.9	37.7	0.008
Location of pancreatic lesion			0.027
Uncinate process	10	1	
Head	40	2	
Body	20	6	
Tail	10	1	
Histological type			0.033
Well	29	1	
Moderate	33	3	
Poor	18	6	
Tumor marker [median (range)]			
CA19-9 (U/mL)	601.5 (2–12,000)	238 (7–6710)	0.634
CEA (ng/mL)	4.6 (0.6–750)	3.3 (2–10.7)	0.351
DUPAN2	365 (25–280,000)	470 (25–4800)	0.57
SPAN1	94 (16–90,000)	150 (5–1950)	0.493
T (T1/T2/T3/T4)	10/11/39/20	1/2/3/4	0.636
N (N0/N1)	48/32	4/6	0.312
M (M0/M1)	44/36	2/8	0.047
Stage (Ⅰ/Ⅱ/Ⅲ/Ⅳ)	10/21/13/36	1/0/1/8	0.163
Treatment			0.327
Operation	23	3	
Chemotherapy	25	1	
Best supportive care	32	6	

^†^ Using the chi-square test, Fisher’s tests, and *t* test for category variables.

## Data Availability

Not applicable.

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
