# Peer review of "Usefulness of Adding Maspin Staining to p53 Staining for EUS-FNA Specimens of Pancreatic Ductal Adenocarcinoma"

_jcm, 2022, doi:10.3390/jcm11206097_

Round 1
Reviewer 1 Report
Comment to the Author
I reviewed your manuscript titled “Usefulness of adding Maspin staining to p53 staining for EUS-FNA specimens of pancreatic ductal adenocarcinoma”. The diagnosis of PDAC on EUS-FNA is sometimes challenging to differentiate from benign disease. Pathological diagnosis using immunostaining is expected to improve the accuracy of differentiation between benign and malignant diseases. However, the commonly used P53 has moderate diagnostic performance in PDAC and is insufficient as an auxiliary diagnosis, so new immunohistochemical staining is needed. The authors focused on Maspin staining and showed its usefulness. Therefore, this paper is informative for the reader in diagnosing EUS-FNA for PDAC. However, some clarifications and corrections would help to better understand and put in perspective your findings:
Comments
1. Are the PDAC patients enrolled in this study only those diagnosed by HE staining of FNA specimens? For example, do the authors include patients who were difficult to diagnose by HE staining but could be diagnosed with PDAC by immunostaining results? This point is very important, and to demonstrate the efficacy of Maspin, patients who could not be diagnosed with pancreatic cancer by HE but could be diagnosed by the addition of immunostaining should also be included.
2. Sensitivity, specificity, accuracy, etc. of HE staining alone should be shown to evaluate the additive effect of immunostaining.
3. I do not understand why the kappa coefficient is used in this study. Please clarify which parameters you are evaluating and why you need the kappa coefficient.
4. The accuracy of immunostaining in FNA specimens needs to be evaluated. It would help if you compare staining differences and concordance rates in immunostaining of resected and FNA specimens, and also validate the cutoff for Maspin staining in FNA specimens.
5. Page 9. Line 301: “the low-expression group of Maspin had significantly more low-grade cancers than the high-expressing group”. Is this sentence correct? It is discrepant from your results.
6. The association between Maspin staining and prognosis is very interesting. Your results indicate a poor prognosis when Maspin is negative. However, Uchinaka et al. have studied using resected specimens and shown the opposite result. It would be best if you discussed the differences in results.
“Cytoplasmic-only Expression of Maspin Predicts Unfavorable Prognosis in Patients With Pancreatic Ductal Adenocarcinoma”
Uchinaka et al. Anticancer Res. 2021 May;41(5):2543-2552. doi:10.21873/anticanres.15032.
Reviewer 2 Report
Comments
The study is very interesting and original.
Methodologically it is well conducted and the statistical part is appropriate.
Therefore, in my opinion, which in any case not being a pathologist I cannot judge the specific part on the merits, I believe that there is only one step to specify better.
The study is retrospective and the pathologist who reviewed the outcome of the immunohistochemical is only one for a period of 5 years ??
That is, there is only one pathologist expert on the subject? Did you do the retrospective review on all selected samples? otherwise, if it was decided before a single pathologist did the review, the study becomes prospective or at least a prospective collection of data has been made on a defined protocol.
Round 2
Reviewer 1 Report
This paper has been well-revised according to my comments.